# Why All Healthy Laboratory Animals Should Be Rehomed, No Matter How Small

**DOI:** 10.3390/ani13172727

**Published:** 2023-08-28

**Authors:** Pascalle L. P. Van Loo, Monique R. E. Janssens

**Affiliations:** 1Animal Welfare Body Utrecht, Utrecht University, 3584 CJ Utrecht, The Netherlands; 2Ethisch Bedrijf (Ethical Company), 3011 XL Rotterdam, The Netherlands; moniquejanssens@ethischbedrijf.nl

**Keywords:** rehoming, adoption, laboratory animals, ethics, chain responsibility

## Abstract

**Simple Summary:**

This paper explores why it is ethical to rehome all healthy laboratory animals after the experiments have finished. We describe our own rehoming experience from the joint Ani-mal Welfare Body of Utrecht University and the University Medical Centre Utrecht, the Nether-lands. During a pilot period, over 350 animals were successfully rehomed. Now, rehoming in our lab is a standard policy and common practice. We discuss several challenges and our responses to those through the continuous evaluation of this adoption program.

**Abstract:**

This paper explores the ethical imperative of rehoming all healthy animals of sentient species after experiments have finished or when they have become otherwise redundant. We take into account disparate perspectives in animal ethics and see how they point in the same direction. We illustrate our case with our own rehoming experience from the joint Animal Welfare Body of Utrecht University and the University Medical Centre Utrecht, the Netherlands. The primary pilot proved successful, after which the principle of rehoming became standing policy and common practice. We discuss several challenges and our responses to those through continuous evaluation of the adoption program.

## 1. Introduction

Rehoming laboratory animals that are alive and healthy after animal experiments, or that have not been used and are surplus, is common practice in animal laboratories. However, this is especially true for larger companion animal species. As normal as it seems to grant cats, dogs and horses a good life after the lab, it seems normal to routinely kill smaller surplus animals such as mice and rats [1,2,3]. This has probably to do with how we were raised: our culture. Many of us have learnt to love dogs, to respect horses, to fear mice, and to be disgusted by rats. The question is whether this discrimination is ethically sound. Note that in some countries dogs are seen as a plague or as food and are treated accordingly. Every year during Dutch Science Weekend, we play a dilemma game about animal experimentation with children aged 8 to 14. Examples of the arguments given for selecting the rat or the mouse from a range of plush animals for their virtual experiments include: “Because there are so many of them”, “I don’t like them very much”, or “My mother is afraid of those.”

One of the Seven Commandments in Georges Orwell’s Animal farm says: “All animals are equal, but some animals are more equal than others”. It expresses how people tend to feel more empathy and compassion towards animals from species with whom they share more traits and to whom they feel closer related [4]. In addition, we recognise these common traits more easily if we encounter animals of the species. They arouse emotions similar to those expressed in human relationships. This explains in part why many of us care more about the fate of non-human primates (who look ‘human’) and dogs and cats (whom we know from close by), while we seem to be less concerned with—literally and figuratively—more remotely living animals, such as farmed animals, rodents, amphibians and reptiles.

The Netherlands National Committee for the protection of animals used for scientific purposes (NCad) was asked by the Dutch Minister of Agriculture, Nature and Food Quality to advise on rehoming of laboratory cats, dogs and non-human primates (NHPs), given the public interest in these species. They state that there is no need for killing laboratory animals. The committee does not take a specific stand towards other animals, but they do ask in their conclusions for “a change of attitude within the field (…), whereby at the end of an experiment animals do not need to be euthanised and can in principle be rehomed, beginning with dogs, cats and NHPs”. One of their proposed actions is: “Encouraging the rehoming of other eligible animal species by means of the general rehoming framework drawn up by the committee”. Which species are seen as eligible by the committee remains unclear [5] (p. 6).

In this paper, we consider whether the option (or even the default) of rehoming healthy laboratory animals after experiments or in case of redundancy should be extended to more animal species. We explore ethical arguments and describe from personal experience a rehoming program of the joint Animal Welfare Body (AWB) of Utrecht University and the University Medical Centre Utrecht in the Netherlands. The program started as a pilot in 2019, focusing on rodents but also open to other species, and was converted to ongoing policy in 2020. It is still up and running. Although the current rehoming policy includes all sentient animals, and a variety of species have been rehomed, the rehoming of rodents was specifically evaluated and fine-tuned several times. By describing our ethical considerations and experiences, we want to contribute to both the academic debate about rehoming and its practical implementation and design in institutions.

## 2. Legislation and Guidelines

The subject of rehoming has been largely absent in laboratory animal legislation and guidelines around the Western world but is gradually being addressed more often. The latest version of the Guide for the Care and Use of Laboratory Animals [6]—which can be considered the most important guideline used in the US and by AAALAC accredited animal facilities around the globe—does not mention rehoming at all. The EU directive 2010/63/EU [7] and Australian Code for the care and use of animals for scientific purposes [8] both promote rehoming. EU legislation states that Member States may allow laboratory animals to be rehomed, and the Australian code states that opportunities to rehome animals should be considered wherever possible. Prerequisites are that the health of the animals allows it, that appropriate measures have been taken to safeguard the wellbeing of the animal (both mentioned in both codes), and that there is no danger to public health, animal health or the environment (EU Directive). Furthermore, establishments should have a rehoming scheme in place that ensures socialisation of the animals that are rehomed (EU Directive).

Regarding species, the EU directive focusses mainly on dogs and cats, stating that these species should be allowed to be rehomed in families as there is a high level of public concern as to the fate of such animals. In general, National Guidelines regarding the rehoming of laboratory animals that are based on legislation primarily focus on dogs and cats [5,9,10]. With this focus, current legislation follows the standing practice that lab dogs and cats have been rehomed for several decades. Many countries have specific organisations that intermediate for dogs and cats (e.g., SHHH in the Netherlands, Jodipro in Belgium, Labor Beagle Hilfe in Germany, Beagles of Burgundy in France).

There is, however, a visible shift regarding species that are considered eligible for rehoming, both bottom-up and top-down. Bottom-up, we see that an increasing amount of research institutes are actively involved in the rehoming of a range of species [11,12]. A recent European survey showed that the 97 respondents had rehomed as many as 23 different species [13]. Top-down, we see that in Germany, the law on Animal Welfare (Tierschutzgesetz) prescribes that killing animals without good reasons is not allowed. This includes laboratory animals and thus implies that researchers need to take into account what happens to the animals they use after the experiment [14]. Furthermore, rehoming has recently been declared an acceptable fate for lab animals by the FDA, based on a bill by Boyle adopted in the US Congress [15]. Since the FDA has a strong influence on the course of many food and drug related animal studies, this declaration paves the way for labs that previously may have been unable to rehome healthy animals. Oddly enough, the Boyle bill specifically excludes mice, rats and birds from being adopted. Although the reason for this is unclear, this may be because rats, mice and birds are still excluded from protection under the US Animal Welfare Act §2132 (g) [16]. Despite this exclusion by the FDA, the European survey shows that rats, mice and domestic fowls are among the species most rehomed [13], emphasising the need for further recommendations and guidelines in this area.

The FELASA working group on Rehoming of animals used for scientific and educational purposes has recently bridged this gap by publishing general recommendations for the rehoming of laboratory animals, including specific recommendations for a number of species [17]. An important statement from this working group is that, although they “planned creating a set of criteria for when an animal can be rehomed or for a suitable adopter”, they now strongly hold the opinion that “it is not appropriate to provide such criteria. The situation should be evaluated on a case-by-case basis, by experts such as veterinarians, animal caretakers and (consulting) animal behaviourists, advised on by the AWB and the Designated Veterinarian” [17] (p. 2).

## 3. Ethical Arguments

Ethics plays an important role in animal testing. From the 1970s onwards, the 3Rs—Replacement, Reduction and Refinement [18]—have step by step gained ground and increasingly formed the ethical and legal basis of all animal experimentation. In the meantime, animal ethics have evolved further and become an indispensable part of both the assessment of project proposals for allowing them a license or not, and daily decision making in the animal facilities and laboratories. Rehoming laboratory animals is one of many much-discussed ethical issues, especially when it comes to smaller animals that are not the common pet species [3].

In animal ethics, it has been argued broadly that animals from many species have a moral status, which means they are morally considerable [19]. In this paper, we focus on vertebrate species, which are all sentient, meaning that animals of these species can experience positive and negative affective states, such as happiness, relief, hunger, thirst, pain and boredom [20]. Additionally, memory, anticipation and the ability to cope with specific circumstances can enhance or diminish negative and positive experiences [21,22]. 

Assuming that redundant laboratory animals will be killed in a relatively friendly way, without unnecessary fear or pain, it is important for our argument that continuing life has, in itself, a positive value for the animal. Or, in other words: is ending a life that could have continued as a good life a harm done to the animal?

The ethics used in animal experimentation is mainly utilitarian [23], based on a harm-benefit analysis: weighing the burdens against the benefits and thus justifying the use of large numbers of laboratory animals for gathering knowledge that can help large numbers of humans or other animals, or protect ecosystems or the environment [24]. The utilitarian case for rehoming all redundant laboratory vertebrates that are healthy and rehomeable is that a life worth living (at least from that moment on) is prolonged, while for the volunteering adopter there is a small burden and a great pleasure. Although current laboratory practice is shaped by the view that the death of an animal is not a welfare issue [2], and that in utilitarianism the topic of the value of continuing life has triggered debate, killing can be seen as taking away future welfare or future fulfilment of preferences [25,26,27,28,29,30].

How is this for other approaches in ethics, such as deontological or rights ethics? Regan argues that animals (at least mammals) have inherent value and that, therefore, their lives matter [31]. Killing them is violating their right to life. Korsgaard argues that other animals than humans are also beings for whom things can be good or bad [32]. They pursue positive experiences just like us humans and are, therefore, ends in themselves that should not be treated solely as instruments for the purposes of others. In addition, there is the deontological case that the animals who have been used for the benefit of humankind deserve a pension-like life in relative happiness. Meijer argues that based on political philosophy, animals should be granted rights [1]. Working animals, such as lab animals, for example, deserve a form of pension or retirement (see also: ref. [33]). A rehoming policy can provide that retirement. The argument in favour of a generous rehoming scheme is also supported by the telos- and virtue-oriented capability approach of animal ethics, that prescribes that we should do justice to animals by allowing them a range of capabilities that offers them the possibility to flourish as the typical beings they are. For all sentient and striving animals, these capabilities include staying alive [34].

Care ethics support rehoming as well. This is an approach that grants feelings of care their own ethical value [2]. In our case, it puts an extra value on the feelings of care of animal technicians, caretakers and researchers who have worked with the animals and taken care of them. They often experience an intimate relationship with the animals, have learned to know them as individuals and feel bothered by having to kill them [35].

Therefore, we take it as a premise that prolonging a life worth living is a value that should be promoted for the sake of the animals. In addition, it makes us more virtuous and caring institutions and staff if we grant animals this opportunity.

If we take the animals’ point of view, it is probable that each of them values their life and their welfare. Nevertheless, it is possible that there are differences between species. It is difficult for us humans to imagine that the life of a mouse equals the life of a horse or the life of our special niece with her great plans for the future. But what are then the criteria for making a difference? Is it recognition that matters, as was suggested in the introduction? Size? Lifespan? Something else? Varner has proposed a continuum: from mere sentience of those who can experience pleasure and pain, via near-personhood of those who have a sense of their own mental states in the past and the future, to full personhood of those who can place their mental states of the past and the future in their life story [22]. On the other hand, it may as well be the case that pain is worse for sentient animals than for persons, who can at least understand for what purpose the pain is inflicted and when it will stop [36,37]. Nussbaum explicitly rejects this type of hierarchical scale [34] (p. 180).

This means that the question how we should value the lives and emotions of different species and individuals against each other cannot easily be answered. On the other hand, there is no need to resolve it here, as in this paper we discuss discrimination as opposed to equal treatment of specifically vertebrate (and often mammal) non-human species. There is no evidence that, for example, chickens and mice occupy an entirely different place on the personhood scale than cats and dogs. The philosopher Meijer herself adopted a group of ten mice, observed them, and described how they showed preferences, communicated and interacted in their small community, and cared for each other [1]. Nevertheless, we will not go into the rehoming of great apes, as they have special statuses in the laws of many countries, great apes being attributed personhood [38]. We conclude that there are no good reasons to discriminate between rats, mice, chickens, dogs, cats, pigs, horses, fishes, frogs, snakes and non-human primates. Why put an effort into rehoming dogs and cats, while routinely killing redundant mice and rats? We should offer all redundant laboratory animals a longer life in good welfare, if possible (see also: [39]). The life of any healthy animal of a sentient species is worth the relatively small effort. 

## 4. Ethical Preconditions

Rehoming is not always the best choice from an ethical point of view. There are preconditions. What is a relevant discrimination criterium is the animal’s chances of a good life in the new home. This is a discrimination criterium on an individual basis that cannot be standardised [17]. At least, the animal’s basic welfare conditions should be guaranteed [5]. This includes the precondition that the animal is appropriately socialised beforehand [39] or can and will be socialised by the adopter. Examples of necessary caution are mentioned by an interviewee in Palmer et al., where there is talk of mice breeds with curly eyelashes that can cause eye irritation or with fast growing teeth that need regular trimming [2].

In addition, public health is a criterium. Rehoming an animal is not imperative when there is a risk to public health [5]. For this reason, laws that do allow rehoming of lab animals exclude genetically altered animals. The purpose of these laws is to ensure that no genetically altered animals unintentionally spread to the wild or end up in the human food chain. Advice against rehoming genetically altered animals is, therefore, given without further debate [17,40]. This rigidity may, however, lead to ethically conflicting situations due to the death of perfectly healthy, adoptable animals, as described in a case by Clark [41]. In this case, healthy, well socialised transgenic pigs were killed even though a suitable home in which the legal requirements were met, was found.

If one takes the ethical imperative to rehome healthy laboratory animal seriously, time and money costs will be made, both to enable the adoption and costs for the animal after adoption has taken place. This can lead to debate about where the responsibility rests. Does it rest with the institution as a whole, the research department, the individual researcher, the AWB, the facility manager, the government, an animal welfare nongovernmental organisation (NGO), or the new owner? Whereas, in most instances, costs after adoption will be for the new owners, we think that at least for the costs to enable adoption, the responsibility lies in the first place with the institution conducting the research. This institutional responsibility can be extended to chain responsibility [28]. In the case of laboratory animals, chain responsibility means that the institution is responsible for a sound origin of the animals (coming from a certified breeder with high animal welfare standards) and a good continuation of their life [23]. The institution can delegate this responsibility on a practical level to the research department, the researcher in charge, the animal facility manager, or the AWB. In that case, high management should make sure that the administrative burden (time) and the financial costs are incorporated in their budgets. This could also be something that institutions include structurally in their funding applications, and funders in their grants. When considering these costs ethically, difficult comparisons have to be made. Although even in utilitarianism, mathematical models for making ethical trade-offs never got off the ground, it is accepted to make this type of rough value comparisons. Doing that, it can be expected that the value a prolonged life in positive welfare has for a sentient individual will outweigh the rehoming costs.

After surgery conducted as part of the experimental set-up, sometimes, restorative surgery is a necessary precondition for an animal to have a good life after rehoming. This can bear difficult ethical dilemmas on those responsible. Cases should be discussed thoroughly during an ethical assessment. Repeated surgery may put a heavy burden on the animal, especially when it is not aware of the purpose of another procedure and the better life lying ahead. In the UK, restorative surgery on laboratory animals for rehoming purposes is not allowed because it is not a permissible scientific purpose [9]. It may be clear that, where there are no legal objections, the case-by-case approach involving several experts as proposed by Ecuer et al. [17] is paramount.

Another issue that should thoroughly be considered is preventive surgery. Some institutions involved in rehoming demand neutering of male animals before rehoming. Although this is an extra burden on both the animal and the institution that wants to rehome the animal, this can be a rule intended to prevent breeding and thus create even more superfluous animals, with indirect negative consequences for other individuals. Although we recognise the argument, we think there are other ways of preventing breeding, such as rehoming of one-sex groups and critical assessment of the adopters. We will discuss this type of practical considerations in the next chapter, where we describe the Utrecht case. 

## 5. The Utrecht Pilot

Rehoming laboratory mice and rats is not a new phenomenon. When we first explored the possibilities for rehoming mice and rats in early 2019, we found on social media several initiatives of individual persons and NGOs committed to rehoming laboratory rats. The Swiss Animal Protection, for example, has a partnership with the University of Zurich since 2018 to enable rehoming of rats. In Poland, graduate student Zosia Pawelska rehomed thousands of lab animals since 2016 with her Lab Rescue initiative. In Wisconsin USA, University Professor Richard Hein has been dedicated for over 20 years to find new homes for each rat that is used by his students. In the past, in our own labs rats and mice have also been rehomed, especially with veterinary students. But this was incidental. In case not enough homes were found, there was no imperative to seek public attention to find extra adopters, and rodents were killed and used in anatomy lessons or discarded.

In 2019, the aforementioned Utrecht AWB, where we both worked, had an annual meeting with Animal Rights, a Dutch–Belgian NGO. During that meeting, a joint ambition came to the table to stop killing healthy animals. We wanted to treat vertebrate surplus laboratory animals equally and include them in the rehoming program as far as possible. We took it upon us to explore things further. We called a meeting with the animal facility manager, a few expert colleagues of the veterinary faculty and a few NGOs: Animal Rights, the Dutch society for the protection of animals ‘Dierenbescherming’ and the experienced rehoming foundation ‘Stichting Hulp en Herplaatsing Huisdieren’ (SHHH). During this meeting we identified some serious concerns. We decided to list the risks and opportunities and find solutions. This resulted in the start of a pilot to find out how to avoid the risks, such as animal welfare risks, rodents being fed to pet snakes, poor matching of numbers of adoptive animals versus adopters and negative media attention; and how to seize opportunities, such as—apart from saving lives—bringing the complex subject of animal experimentation closer to the general public and making it more transparent.

The SHHH left the working group after giving useful advice. The others became partners in the pilot. To keep things simple, we decided to work with closed stock: each partner would do their part. The animal facility would accommodate the animals until adoption was arranged (maximum two weeks), the AWB took responsibility for the coordination and the communication, and Animal Rights volunteered to do the transportation. Together with the team of the AWB Animal Welfare Officers we decided upon the preliminary conditions for animals and adopters. Animal Rights found a rodent shelter (Het Knagertje) that was willing to take in animals that could not be rehomed immediately, as a buffer. They would—like they were used to—charge the adopters a small amount of money as compensation (partly) for the shelter costs and also as a negative incentive against buying the rodents as food for other animals. Animal Rights posted a first call for adopters on their Facebook page. The first groups of rats and mice were put up for adoption and were rehomed. The pilot ran for half a year. During this period 280 mice and 75 rats were adopted as well as a small group of zebra finches. We became so enthusiastic that we took up the challenge to find a solution for all adoptable animal species that we encountered in our facilities. 

## 6. Current Utrecht Rehoming Policy and Practice

Our experience with the pilot was very positive: contact between the parties involved was pleasant, we had ample potential adopters—even leading to a waiting list at some point—and media exposure (including reactions) was very positive. Furthermore, animal care staff and researchers involved were, and still are, very enthusiastic about the possibility to rehome ‘their’ lab animals. They gladly commit the extra time needed to socialise, pack and dispatch the animals. As one of our animal technicians explained: “Rats and mice are very social and smart animals. When you are an animal caretaker or technician there is nothing nicer than to know that your animal, if it is still healthy, will get a good home after the study. I am quite willing to devote some time into that.”

From the start of the pilot in October 2019 to the end of 2022, over 500 mice, 1000 rats and 240 animals of other species have been rehomed. Other species include chickens, guinea pigs, hamsters, goats, zebra finches and even a python and an iguana (Figure 1).

Despite our initial positive experience, there were also some concerns regarding animal welfare and legislation, and we have, therefore, built a framework to ensure that we abide by the law, that the animals we rehome are suitable for adoption and will properly be taken care of in their new homes. The main principles we follow for the animals largely correspond with the general recommendations later described by Ecuer et al. [17] as well as with the species-specific recommendations they provide [42], which implies that these are fairly intuitive:Reuse of laboratory animals aids in reducing the total number of animals used. Therefore, before animals are put up for adoption, reuse is considered;We rehome healthy wild type animals only. Genetically modified animals are legally not allowed to leave our lab;We rehome animals that are socially capable of adjusting to their new home and are social towards human beings. Mauri and Bonelli found that former laboratory rats kept in a social group establish complex social structures and interact preferably with litter mates and strain mates [43]. Their social behaviour is restructured when new individuals are introduced. Likewise, we generally rehome social species in existing stable social groups of at least two animals. This is always the case for rats, mice and chickens. For other social species, single rehoming may be considered if introduction into the new home and to other pets is carefully planned. Due to the high risk of aggression, we exclude male mice from rehoming. This issue is up for discussion, since the shelter we work with claims to have very good experience with housing and rehoming groups of male mice;We rehome animals that have an acceptable life expectancy (i.e., young to middle-aged) and are healthy at the moment of rehoming;We ask adopters to sign a rehoming agreement in which ownership of the animals is transferred. Amongst others, this agreement entails that the new owners take full (also financial) responsibility for adequate housing and care, including veterinary care. The new owners are not allowed to use the animals for breeding.

We provide adoptive owners with the following list of recommendations:


Respect the natural behaviour and species-specific needs of the animals, e.g., take note of the nocturnal nature of mice and rats;Follow the advice on housing and husbandry of pet animals from the National Companion Animal Information Centre (LICG) https://www.licg.nl/, accessed on 9 August 2023;Keep the animals in their existing social groups or, if applicable, mix groups very carefully;Provide the animals with a large living space with ample enrichment;Avoid confrontation with predatory animal species;Perform regular health checks and visit a vet when necessary.


Before the animals are packed for transport, they receive a thorough health check by a veterinarian or an animal caretaker. They also get some food packed with them to ensure a smooth transition in diet [17] and (for rats and mice) pieces of cucumber as a small water supply, even though car transport in the Netherlands from the centrally located city of Utrecht hardly ever lasts longer than two hours.

Personal contact with the new owners and exchange of photos or film material before and after the adoption process ensures that we leave the animals in adequate hands (Figure 2 and Figure 3).

## 7. Follow up and Feedback

Since many of our mice and rats are rehomed through animal shelter ‘Het Knagertje’, this shelter is audited by us to ensure that all animals are well cared for, that a veterinarian visits regularly and that the adoption process when animals are adopted to private homes is professionally set up. We see this as part of our chain responsibility. From these audits, we learned that it is easier to find rehoming addresses for mice than for rats, which is in concordance with Franco and Olsson, who state that adopting mice takes little effort [23], as the animals are small and easy to accommodate. In addition, mice are often rehomed in groups of 5 to 10 animals, while rats are usually rehomed in groups of 2 or 3. Also noteworthy to mention is that, according to the shelter caretakers, our rats and mice are very healthy and well socialised compared to other rescued rodents that the shelter receives.

The primary concern arising from these audits is that the shelter is very crowded, and it may be a challenge to find enough suitable rehoming addresses, especially if a large number of animals become available for adoption at once. Recent data have shown that not all our animals find a new home and, hence, stay in the shelter until they die of natural causes. A recent enquiry revealed that from 57 mature rats brought to the animal shelter, 30 were still living there after 5 months, with little chance of being chosen for adoption due to their age. On average, 15–20 percent of our rats never leave the shelter, whereas all our mice find new adoptive homes. Our assessment is that living in a shelter is preferential over euthanasia in the laboratory. The rats usually live in spacious cages in fairly large groups. They receive a lot of cage enrichment and seem to be living a good life, not too different from a life in adopters’ homes. Only the interaction with people is considerably less. However, we think that the animals receive enough attention from each other. The ethical consideration that continuing their lives has value for these animals is the deciding factor for us. Nevertheless, we devote lots of energy to finding more rehoming addresses, using the following channels:Animals are advertised on the online marketplace for adoption run by the Dutch Animal Protection Society. This has been very helpful to keep the numbers of rehoming addresses up to par after the first wave of volunteers were provided with animals;We regularly advertise on Utrecht University intranet and in newsletters from Utrecht student societies;We send direct mailings to people who have previously shown interest in adoption. These include petting zoos and ‘green schools’ that educate animal caretakers. With the latter, we ensure that our animals are only used for teaching animal care and handling.

## 8. Questionnaire Results

We are very open to comments and questions from people who adopted our animals, and we regularly send questionnaires to the private homes to keep track of how our animals are doing. By doing so, we can build a database of information on how the animals grow old in the private homes and adapt our adoption process if necessary. 

Since the start of the pilot project, we sent out 4 questionnaires, generating a total of 147 responses. Evidently, some responses originated from the same respondents over time. In total we have received 128 unique responses, covering 231 rats, 193 mice, 94 chickens and 10 other animals. This is 28% of the total number of animals we rehomed. The majority of respondents (80%) have an adult-only household. Most households have other pets, mostly other rodents (25%), dogs (19%) or cats (18%). Twenty-seven percent do not have any other pets.

Over 95% of the respondents were satisfied with the adoption process and the information they received. With regard to adopted rodents, we asked about the level of interaction and the approachability of the animals. In Table 1, we show the interaction between these two aspects. Although we cannot distil a clear trend from Table 1, we do see that with multiple interaction moments each day, more animals approach their caregivers spontaneously or are well trained.

A total of 66 out of 528 animals were reported to have died. Table 2 shows the possible causes of death. For rodents, a range of health issues were reported, with lung problems (13 incidents) and skin problems (8 incidents) being mentioned most. Other health issues mentioned were eye, ear and paw infection; tumour development; elephant’s teeth; and aggression.

In addition to the statistics from the questionnaire, the added comments provided by some of the adopters are especially valuable. These pointed out the need for additional information prior to adoption. Respondents request information on general animal characteristics such as light sensitivity of albino animals, on the procedures to which the animals have been subjected and on the individual’s character.

## 9. Lessons Learned, Ethical Refinement of the Procedure

Having a follow-up system in place is something that current recommendations advise [17], and we strongly agree with this. Our follow-up questionnaires have provided us with much detailed, worthy information to refine our adoption program and increase our adoption success. It has also helped in building bonds with several new owners who turned out to become regular adopters and ambassadors to our program.

The reported health issues and causes of death, although not worryingly high, revealed two phenomena that needed our attention: On three occasions, lung infections were reported in animals that were mixed with other animals not from our lab. Although the incidence of such reports is still relatively low, we suspect that the naïve immune system of lab animals makes them more susceptible to pathogens that are carried by more robust pet store animals. Secondly, the number of deaths with unknown cause was relatively high, implying that recognising signs of ill health may be difficult for some adopters.

With regard to the animals’ characteristics, we learned that it is important to manage expectations of adopters and give them guidance on habituation and training on the one hand and assess the socialisation of the animals when they are still with us on the other hand.

The feed-back of the questionnaires has thus led to the following changes in our adoption process: Prior to adoption, each animal’s behaviour is evaluated and declared fit for adoption. If necessary, extra time is invested in socialisation and habituation to larger housing;We updated the health certificate to include common characteristics such as sensitivity to light of albino animals and susceptibility to pathogens, as well as individual behavioural characteristics where applicable;We produced a short information video in which we explain how to recognise the most common health problems in rats and mice;In case animals are mixed with other animals, we point out the animals’ naïve immune system and discuss whether the adoption can proceed. If so, we point out a good online instruction manual on how to introduce new animals in an existing social group;We publish all project licenses for animal experiments on our website; therefore, we can provide the adopters with specific background of the project in which the animals have been used;We are currently creating a short instruction video on how to acclimatise rats and mice, and how to train and interact with them, with special reference to interaction involving young children.

The majority of questionnaire responses (77%) were returned from adopters who adopted directly from Utrecht University. To receive responses from people who adopt through Knagertje or Animal Rights, we depend on those organisations as mediator. Apparently, this only yields a small number of responses. We were, therefore, not able to research trends in, for example, sociability or disease-susceptibility of the animals when a shelter is used as intermediate.

## 10. Media

One of the concerns of colleagues, partner organisations, and organisations that showed interest emulating our project was fear of negative media exposure. This is also reported by Skidmore as one of the reasons of some facilities for not engaging in rehoming [12]. Especially for organisations not used to going public with their involvement in animal experiments, media exposure related to this topic can feel uncomfortable. At the same time, the Utrecht AWB and our predecessors in Utrecht University and the University Medical Centre Utrecht already had a long history of transparency around animal experimenting and positive contact with the media. We regularly welcome journalists and TV crews in our laboratories and explain what we do there, why and how. What is helpful is that we have among our AWB staff a dedicated and specialised communications consultant.

We wrote a factual and honest article for our website that no doubt reflected our own enthusiasm for the project. In 2003, Carbone et al. already pointed out the importance of a well-written press release when engaging in an adoption program for lab animals [40]. We also prepared a Questions and Answers document as preparation for critical questions. Then we posted messages and pictures on social media, especially with the goal of finding homes for the animals. While there was some coordination, we obviously had less control over the posts of our partner NGOs. However, everyone was driven to make the project a collaborative success. Even though at that point we did not actively seek the media (we were still conducting a pilot), the story was picked up by several newspapers. We were approached by journalists for more information. In most cases, they wanted to talk to the people who had adopted animals. After asking for their consent, we brought them into contact with the journalists. Many success cases of happy people and happy animals were reported in the media. We received many positive reactions on social media. To be even better prepared for media questions, we have now put together an information set that we can easily share with journalists and, when necessary, we adapt the Questions and Answers document.

## 11. Discussion and Conclusions

Ethical reflection shows that rehoming all healthy sentient animals, of which a reasonable period of a good life in a positive welfare status can be expected, is the right thing to do. Their life is of great importance to the animals concerned (without their life, they have nothing); therefore, costs and time burdens on the facility will not easily outweigh the value of proceeded life to the animal concerned. Therefore, making the effort to rehome animals is imperative. At least all vertebrates (who are all sentient animals) should receive equal consideration and effort. This is part of the chain responsibility of those—both people and institutions—who use animals for experimenting. It is their responsibility to see to the welfare of the animals from birth to death, and therefore also to the prolonging of their life in good conditions.

The Utrecht case shows that a rehoming program can be implemented and executed successfully. Despite the extra workload, animal care staff and researchers happily invest time in socialising the animals, and in packing and dispatching them for adoption. This, in our view, is a shining example of culture of care in which animal welfare and job satisfaction go hand-in-hand. LaFollette et al. correlated euthanasia of lab animals with compassion fatigue in animal care staff, and animal welfare initiatives with compassion satisfaction [35]. An adoption program such as ours cuts on both these ends, decreasing the need for euthanasia and replacing it with an initiative that greatly improves the total quality of life of the animals. 

Our case also shows that constant evaluation is necessary. We learn on a daily basis from experience and feedback. Very recently, rats have been rehomed that appeared to be under-socialised. Some adoptive people managed to socialise them with a lot of patience, others were disappointed about not being able to interact with the rats as they had hoped to. Two rats were returned to Animal Rights for this reason. We reacted to this issue by tightening up our internal socialisation program and our adoption criteria as well as adding to our videos a new one on how to build up interaction with rats. A learning point was that we need to ensure that adopters have realistic expectations [40].

Rehoming animals means that they become pets, which means that they change categories. Decisions about veterinary treatment will be in the hands of their adopters and, therefore, become less consistent than in the lab [2]. This bias can go both ways: more or less treatment, longer or shorter prolonging of life. This does not have to be problematic, as long as the bias remains between an acceptable range. Nevertheless, just like for all pets, this is a risk to animal welfare that can be controlled by thorough assessment of adopters and by providing relevant and easily accessible information, such as short videos.

Most of the feedback we receive stems from direct adopters. Carbone et al. compared advantages and disadvantages of direct versus indirect adoption [40]. Indirect adoption has the advantage that time investment is lower, potential adopters can visit the animals prior to making a decision, shelter staff can help with socialising animals further and may take back animals when adoption is unsuccessful. Direct adoption, on the other hand, enables you to have close contact with the adopter to ensure that there is a good match. We have experience with both. Direct adoption takes more effort, but the interaction with the adopters and feedback we receive enables us to improve upon our program. Since the shelter is very full, we feel hesitant to bring many animals there. A solution that combines the best of both worlds, a sanctuary within our facility specifically aimed at laboratory animals, is an idea we are currently exploring.

Preventing redundance of laboratory animals by, for example, using both sexes of a species and efficient planning go first [23], but if there are healthy animals that cannot be reused for practical or ethical reasons (putting too much burden on the same animal), rehoming is morally obligatory, as was argued above. With joint effort, animals can move from being a scientific instrument to being a loved family member [12]. This practice of rehoming animals more consequently and enabling them to move from scientific instruments in a laboratory context to loved pet animals in a home context could even play a role in reshaping the moral landscape and the way society sees and treats animals in general [12,44].

Other institutions, both in the Netherlands and abroad, have shown interest in our rehoming program. We actively share our experiences, both in one-to-one meetings and congresses. During the 2022 Federation of European Laboratory Animal Science Associations (FELASA) congress, we hosted a double session on rehoming where we shared experiences with several other institutions and an interested audience [45,46]. We can only hope that more and more institutions dealing with laboratory animals will follow and start saving all the lives they can.

We realise that there is an end to the number of possible homes, especially for rodents, although we think there is plenty of potential for these easy to keep pets, that have relatively short lives. We cannot be sure what would happen if rehoming would become default in the full animal experimenting community. For this reason, and also because much larger numbers of animals are not suitable for rehoming, we think that apart from this Refining measure, having an adoption scheme does not relieve anyone of the responsibility of continuing to seek for Replacement and Reduction alternatives. 

Until now, reactions of all people involved have been overwhelmingly positive, both inside our lab from animal care takers and researchers who gladly walk the extra mile to get the animals adopted, and from the animal advocates and new homes who thank us for allowing these animals a second life. And last but not least, we think we can be sure that the more than 1700 animals that have been rehomed, until this day, lead or have led a happy life.

## Figures and Tables

**Figure 1 animals-13-02727-f001:**
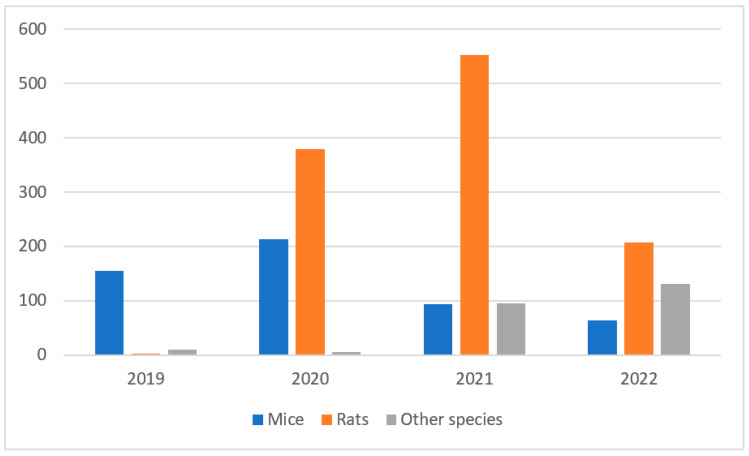
Mice, rats and other species rehomed since 2019.

**Figure 2 animals-13-02727-f002:**
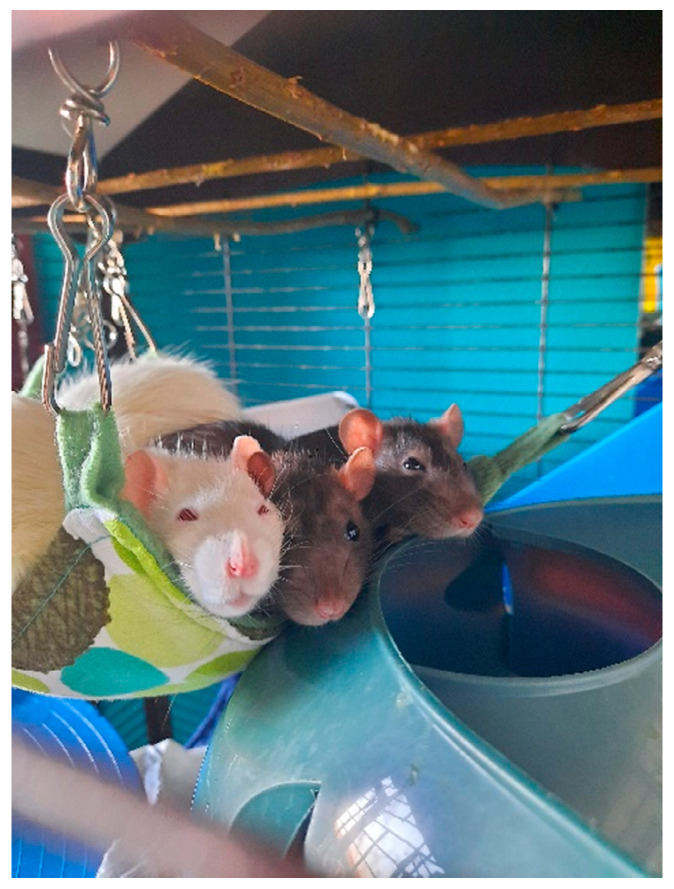
Three adopted rats in their new home. Photo: Julia van Eupen.

**Figure 3 animals-13-02727-f003:**
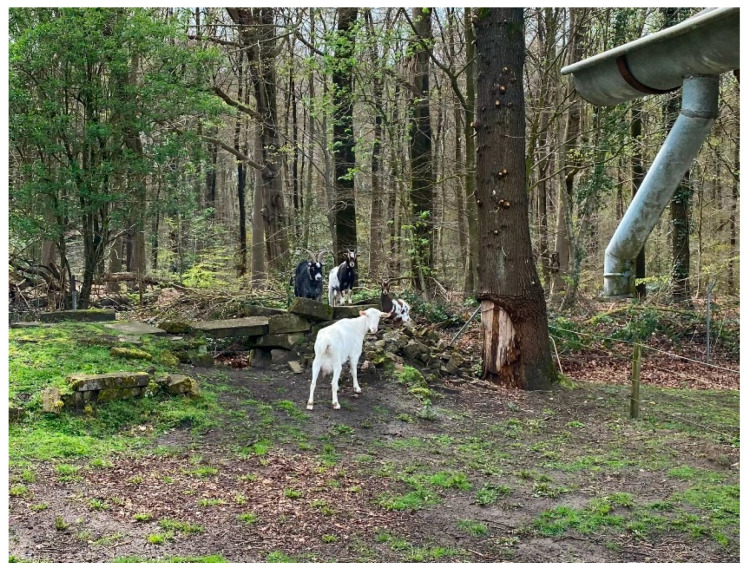
An adopted Saanen goat meets her new friends. Photo: Tamara Dorresteijn.

**Table 1 animals-13-02727-t001:** Amount of time spent with animals versus animal attitude. Respondents were asked how often they interacted with their animals (options: necessary care/(almost) daily interaction/interaction multiple times a day) as well as how approachable their animals were (options: animals are scared of humans/animals accept human approach/animals actively approach humans/animals are well trained and interactive).

	Scared	Accept	Approach	Trained	Total
Necessary care	3	23	20	0	**46**
(Almost) daily	1	13	33	7	**54**
Multiple times	0	4	24	7	**35**
**Total**	**4**	**40**	**77**	**14**	

**Table 2 animals-13-02727-t002:** Animals reported to have died or euthanised with their probable cause.

	Total	Unknown	Old Age ^1^	Pneumonia	Heat Shock	Heart
Mice	18	6	11	1	1	
Rats	36	8	21	6		1
Hamsters	5	3	2			
Chicken	7	3	3	1		

^1^ Old age was often mentioned in combination with tumours. Tumours were not mentioned as single cause of death or single reason for euthanasia.

## Data Availability

Publicly available datasets were analysed in this study. This data can be found here: https://ivd-utrecht.nl/nl/infocentrum/document/vragenlijst-adoptie-enquete, accessed on 9 August 2023 (questionnaire) and https://ivd-utrecht.nl/nl/infocentrum/document/ruwe-data-enquete-adoptie, accessed on 9 August 2023 (responses).

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
