# Peer review of "Why All Healthy Laboratory Animals Should Be Rehomed, No Matter How Small"

_animals, 2023, doi:10.3390/ani13172727_

Round 1
Reviewer 1 Report
I found the article interesting; however, I think if the 3 Rs are followed and planned correctly, the problem of the use and surplus of not-used animals could be solved.
Author Response
Dear reviewer 1,
thank you very much for your positive feedback on our paper. You have made one comment: …if the 3 Rs are followed and planned correctly, the problem of the use and surplus of not-used animals could be solved.
If we understand this comment correctly, we agree with you that the first aim should be to reduce the existence of surplus animals to an absolute minimum. We have several policies in place to ensure an optimal matching of supply and demand of animals used in our facility, e.g.:
https://ivd-utrecht.nl/en/infocentre/document/policy-on-purchasing-and-breeding-laboratory-animals and https://ivd-utrecht.nl/en/infocentre/document/policy-on-surplus-laboratory-animals-reuse-and-re-homing
However, even when matching is optimal, we do have surplus animals. For example, rats that have been used in experiments, are not fit for reuse in other experiments, but are otherwise young, sociable and healthy.
Please note that, even though our rehoming program is successful, our first and foremost aim is to replace and reduce the number of animals used as we have stated in our discussion.
Reviewer 2 Report
This is an excellent and informative piece. The only question I would have is how the animal shelter feels about having to keep hold of the rats because they were not adopted. Is it possible they will stop accepting rats from the labs?
Author Response
Dear reviewer 2,
Thank you very much for your positive feedback on our paper. You ask ‘how the animal shelter feels about having to keep hold of the rats because they were not adopted. Is it possible they will stop accepting rats from the labs?’
Your question is indeed a concern. At this time, only a few animal facilities in The Netherlands are starting with a rehoming program according to our example. If this number increases, the number of adoptive homes may become a limiting factor. The shelter we work with, has a relatively large buffer to accept animals. Nevertheless, we are broadening our scope to involve more shelters and, as we mention in our paper, we are constantly reviewing our policy to ensure the reduction of animals and finding enough suitable homes for our animals.
Reviewer 3 Report
Dear authors,
thank you for this great idea and the option to share it with a greater readership than only in dutch media! The combinatin of both ethical background, description of the project and results is great. I immediatly was thinking about how we can do it in our institution.
Although I recommend (for the first time in my scientific life) a direct publication, I have 4 mini-points which just were a amall issue in my enthusiastic reading. perhaps there is an option to consider these:
L. 118 A ' is missing at the end of the phrase.
L 277 The different initiatives had links but not the NGO "animal rights", perhaps worth to include
L 569 "an" instead of "and" ?
L570 there is too much space between two words.
Thanks for sharing your ideas and your enthusiasm!
Author Response
Dear reviewer 3,
Thank you very much for your positive feedback on our paper. We have amended the editorials you mentioned. Please feel free to contact us if you would like to implement this in your institution.